# Plastid Genome Evolution of Two Colony-Forming Benthic *Ochrosphaera neapolitana* Strains (Coccolithales, Haptophyta)

**DOI:** 10.3390/ijms241310485

**Published:** 2023-06-22

**Authors:** Ji-San Ha, Duckhyun Lhee, Robert A. Andersen, Barbara Melkonian, Michael Melkonian, Hwan Su Yoon

**Affiliations:** 1Department of Biological Sciences, Sungkyunkwan University, Suwon 16419, Republic of Korea; yama4507@gmail.com (J.-S.H.); duckhyunlhee@gmail.com (D.L.); 2Friday Harbor Laboratories, University of Washington, Friday Harbor, WA 98250, USA; raa48@uw.edu; 3Group Integrative Bioinformatics, Department of Plant Microbe Interactions, Max Planck Institute for Plant Breeding Research, 50829 Cologne, Germany; bmelkonian@mpipz.mpg.de (B.M.); mmelkonian@mpipz.mpg.de (M.M.); 4Central Collection of Algal Cultures (CCAC), Faculty of Biology, University of Duisburg-Essen, 45141 Essen, Germany

**Keywords:** Coccolithales, plastid genome, *Ochrosphaera*, coccolith, LIPOR genes

## Abstract

Coccolithophores are well-known haptophytes that produce small calcium carbonate coccoliths, which in turn contribute to carbon sequestration in the marine environment. Despite their important ecological role, only two of eleven haptophyte plastid genomes are from coccolithophores, and those two belong to the order Isochrysidales. Here, we report the plastid genomes of two strains of *Ochrosphaera neapolitana* (Coccolithales) from Spain (CCAC 3688 B) and the USA (A15,280). The newly constructed plastid genomes are the largest in size (116,906 bp and 113,686 bp, respectively) among all the available haptophyte plastid genomes, primarily due to the increased intergenic regions. These two plastid genomes possess a conventional quadripartite structure with a long single copy and short single copy separated by two inverted ribosomal repeats. These two plastid genomes share 110 core genes, six rRNAs, and 29 tRNAs, but CCAC 3688 B has an additional CDS (*ycf55*) and one tRNA (*trnL-UAG*). Two large insertions at the intergenic regions (2 kb insertion between *ycf35* and *ycf45*; 0.5 kb insertion in the middle of *trnM* and *trnY*) were detected in the strain CCAC 3688 B. We found the genes of light-independent protochlorophyllide oxidoreductase (*chlB*, *chlN*, and *chlL*), which convert protochlorophyllide to chlorophyllide during chlorophyll biosynthesis, in the plastid genomes of *O. neapolitana* as well as in other benthic Isochrysidales and Coccolithales species, putatively suggesting an evolutionary adaptation to benthic habitats.

## 1. Introduction

Haptophytes, along with cryptophytes, alveolates, and stramenopiles, belong to the group of taxa that underwent secondary endosymbiosis (CASH lineage), possessing a red algal-derived chlorophyll-*a,c* containing plastid [1,2], and they thrive predominantly in marine environments. Haptophytes are named for the haptonema, a flagella-like filamentous appendage involved in prey capture and cell attachment. Marine haptophytes contribute up to 10% of global carbon cycling through fixing CO_2_ by photosynthesis and biomineralization [3,4]. Furthermore, they account for 50% of the calcium carbonate precipitation in the oceans [5,6].

Approximately 80 extant genera and 330 species of haptophytes have been described; however, clone libraries of environmental samples have revealed an additional hidden diversity [7]. Haptophytes are classified into three classes: Coccolithophyceae (=Prymensiophyceae; equal homodynamic flagella and plate scales), Pavlovophyceae (unequal heterodynamic flagella and knob scales), and the recently added Rappephyceae (equal heterodynamic flagella and knob scales) [7,8,9,10,11]. The Coccolithophyceae are subdivided into six orders; two orders (Phaeocystales, Prymnesiales) lack mineralized scales or coccoliths, whereas four orders (Isochrysidales, Syracosphaerales, Zygodiscales, and Coccolithales) have coccoliths [12,13]. Most of the Coccolithophyceae are known from the plankton, but at least some species form benthic colonies as part of their lifecycle [14]. The benthic stage consists of attached palmelloid cell masses, where the cells are without flagella and surrounded by cell walls. This stage alternates with the planktonic stage of flagellate cells (e.g., *Chrysotila* and *Ruttnera* in [14,15]). Because of the distinct heteromorphic features of these two stages, the stages were recognized as two distinct species (or genera) until laboratory culture experiments connected the two life stages as part of the lifecycle of one organism [16].

The genus *Ochrosphaera*, classified in the Coccolithales, is frequently found in various benthic coastal sites of the Mediterranean Sea as well as coastal areas of the Northern Atlantic, Indian, and Pacific Oceans [17,18,19]. *Ochrosphaera* cells lack a haptonema, they are covered with coccoliths, and they have two thin parietal golden-yellow plastids [18,19]. *Ochrosphaera* forms two types of coccoliths (vase-shaped and pully-shaped), and coccolith morphology is used as one of the key characteristics for species identification [20].

Chlorophyllide is a precursor to chlorophyll-*a* and -*b* and is converted from protochlorophyllide by enzymes of POR (light-dependent protochlorophyllide oxidoreductase) or LIPOR (light-independent protochlorophyllide oxidoreductase). Accumulation of protochlorophyllide is harmful to a wide range of organisms, including growth inhibition in cyanobacteria [21] and toxicity in plants [22]. The POR enzyme is known to convert protochlorophyllide to chlorophyllide in the presence of light, with blue light being three to seven times more effective than red light for POR activation. LIPOR can also convert protochlorophyllide independent of light conditions, but LIPOR proteins are apparently sensitive to oxygen due to the possession of iron–sulfur clusters such as nitrogenase [23,24,25,26]. LIPOR is comprised of three subunit genes of *chlB*, *chlN*, and *chlL,* and it has been reported from cyanobacteria, Viridiplantae, red and glaucophyte algae, and red algal plastid descendants (cryptophytes, alveolates, stramenopiles); however, LIPOR genes have not been previously reported for haptophytes and euglenophytes [27,28,29].

Until now, six nuclear genomes and 11 plastid genomes are reported for haptophytes but the coccolithophores are represented by only two Isochrysidales plastid genomes [30,31,32]. In this study, we generated and analyzed plastid genomes from two strains of *Ochrosphaera neapolitana* that were collected from the Canary Islands, Spain, and Treasure Island, Florida, USA. These two genomes were used to test infraspecific variation. The plastid genomes were compared with all the available haptophyte plastid genomes, as well as other chlorophyll-*c* containing sister taxa (i.e., cryptophytes and stramenopiles) and their plastid donor, red algae. We unexpectedly found the LIPOR genes only in the plastid genomes of some Isochrysidales and Coccolithales species including the two *O. neapolitana* strains. Our results provide a better understanding of the evolutionary history of the haptophyte plastid genome.

## 2. Results and Discussion

### 2.1. Phylogenetic Relationships and Morphological Characteristics of Ochrosphaera neapolitana

Phylogenetic analysis of 18S rRNA from 77 haptophyte taxa revealed that the Gran Canaria, Canary Islands, Spain (CCAC 3688 B) and Treasure Island, FL, USA (A15,280) isolates were grouped together with other previously reported 18S rRNA sequences from 15 *Ochrosphaera* strains (bootstrap support, BS 61%, see Figure 1). The monophyletic *Ochrosphaera* clade was grouped with the *Hymenomonas* clade including *H. coronata* ALGO HAP58 and *H. globosa* ALGO HAP30 (BS 97%, sequence divergence between two clades = 0.007–0.017); these were followed by a sister-group relationship with the *Chrysotila* clade (BS 83%). 

Within the *Ochrosphaera* clade of the 18S rRNA tree (Figure 1A), two distinct clades were recognized (sequence divergence between two clades = 0.003–0.007 when the short JF708125.1 sequence was excluded). Strain CCAC 3688 B was closely related to CCAP 932/1 and RCC2959 (BS 96%, sequence divergence = 0–0.004), whereas strains A15,280, NIES-1964, NIES-1395, and UTEX LB 1722 were grouped with eight published 18S rRNA sequences (BS 97%, sequence divergence = 0–0.001). Sequences from GenBank were identified as *O. neapolitana*, *O. verrucosa*, and *Ochrosphaera* sp. 

Within the 28S rRNA tree (Figure 1B, Appendix A), the monophyletic *Ochrosphaera* clade was grouped with the *Hymenomonas* clade, the same as 18S rRNA tree. The 28S rRNA tree also showed that the *Ochrosphaera* clade was subdivided into two clades, with CCAC 3688 B in one clade and A15,280 in the other clade (Figure 1B). 

Morphologically, the cells of CCAC 3688 B and A15,280 each had two parietal golden-yellow plastids with pyrenoids; however, there were some morphological differences between strain A15,280 and the other *Ochrosphaera* strains (Figure 2). For example, A15,280 had a smaller cell size of 3.96 ± 1.36 µm (*n* = 20) than CCAC 3688 B (4.84 ± 0.85 µm, *n* = 20) and other strains [NIES-1964: 4.86 ± 0.70 µm (*n* = 20); NIES-1395: 4.88 ± 0.61 µm (*n* = 20)] (Figure 2). Under the same cultivation conditions, A15,280 produced naked cells that were covered with mucilage (Figure 2D–F), while the other strains typically formed colonies whose cells were covered with coccoliths (Figure 2A–C, G–L). 

*Ochrosphaera neapolitana* was originally described using isolates collected from the coast of Naples, Italy [33]. Two additional species, *O. verrucosa* and *O. rovignensis*, were described by the same author using samples collected from the Adriatic Sea coastline in what is now Croatia [34]. The three species were separated based on cell sizes (*O. neapolitana*; 4.6–6 μm, *O. verrucosa*; 8–11.5 μm, and *O. rovignensis*; 11–13 μm) [20]. Thus, all the *Ochrosphaera* strains in this study fit the cell size for *O. neopolitana*, the type species. 

The SEM observations showed that strains CCAC 3688 B, NIES-1395, and NIES-1964 contained both pully-shaped and vase-shaped coccoliths with six to eight elements (Figure 2). These two types of coccoliths were not detected for A15,280, which produced incomplete coccoliths (Figure 2E,F). Gayral and Fresnel-Morange (1971) studied *O. neapolitana* based on samples from Luc-sur-Mer Marine Station, France, and they suggested that the vase-shaped and pully-shaped coccoliths were key characteristics for the type species [19,20].

Molecular phylogenetic studies showed that A15,280 grouped with NIES-1395, NIES-1964, UTEX LB 1722, and other strains in both the 18S and 28S rRNA trees, and we concluded that all four strains should be identified as *O. neopolitana*. Therefore, we assume that A15,280 secondarily lost its ability to form complete coccoliths.

### 2.2. General Features of the Plastid Genomes 

The complete circular plastid genomes of *O. neapolitana* CCAC 3688 B and A15,280 had assembled lengths of 116.9 kb and 113.6 kb, respectively (Figure 3A). They had a quadripartite structure containing a long single copy (LSC) and a short single copy (SSC) separated by two ribosomal inverted repeats (IRa, IRb). The genomes shared a core set of 110 protein-coding genes, six rRNAs and 29 tRNAs, but strain CCAC 3688 B had an additional coding sequence (CDS) (*ycf55*) and one tRNA (*trnL-UAG*). These two gene deletions affected the synteny block difference, i.e., one inversion *(trnF* to *ycf55*) and one gene transposition (*ycf45*) that distinguished the two plastid genomes (Figure 3A). High divergence rates were found between the genomes in the intergenic regions (IGR), and this was due to numerous short sequence insertions and deletions (indels) as well as single nucleotide polymorphisms (SNPs) (Figure 3B). Especially, two large insertions at the IGRs (2 kb insertion between *ycf35* and *ycf45*; 0.5 kb insertion in the middle of *trnM* and *trnY*) were detected, and these large insertions at the IGRs and the two additional genes resulted in the CCAC 3688 B plastid genome (116.9 kb) being 3 kb larger than that of A15,280 (113.7 kb) (Figure 3). 

Divergence rates between two strains of *O. neapolitana* were calculated based on the DNA sequence alignment (Figure 3B). The average DNA sequence variation between the two strains was 25.2%, while that of only the genic region was 11.9%. Divergence rates higher than 30% were mostly found in IGR and two transposition regions; however, one exceptional genic region case was found in *ycf80* (39.7%). These genetic features, along with incomplete coccolith formation in A15,280, represent infraspecific variation of *O. neapolitana*. 

The two new *O. neapolitana* plastid genomes and eleven publicly available haptophyte plastid genomes were compared (Figure 4). The genome sizes ranged from 95.281 kb (*Diacronema lutheri*) to 116.9 kb (*O. neapolitana* CCAC 3688 B) (Figure 4A). The number of protein coding genes ranged from 108 to 119, the GC content ranged between 35.4 and 36.9%, and no introns were found in the plastid genomes. The average size of the IGR ranged between 89.4 and 143.4 bp, but the two *O. neapolitana* plastid genomes fell outside this range, and the IGRs measured 167.6–176.1 bp in size (Table 1). Among the plastid genomes of rhodophytes and the CASH lineage (cryptophytes, alveolates, stramenopiles, and haptophytes), the cumulative IGRs sizes were largely correlated with the plastid genome size (Appendix A). In the case of the haptophytes, the two *O. neapolitana* plastid genomes contained more than 20% IGRs (21.1–22.0%), whereas other species with smaller genomes had smaller portions of IGRs (13.2–18.9%), therefore supporting the hypothesis that IGR size is correlated with genome size. 

Although there was a conserved plastid genome structure within each order of haptophytes, low structural integrities were found between the orders, especially in the copy number of ribosomal repeats and their constituents (tRNAs and rRNAs) (Figure 4B,C). The Pavlovales had one copy of the ribosomal operon, which was formed by *16S rRNA*-*trnL*-*trnA*-*26S rRNA,* but the other haptophyte taxa (i.e., Rappephyceae and Coccolithophyceae) had a pair of inverted repeats that created a quadripartite structure in the genome. *Pavlomulina ranunculiformis* (Pavlomulinales) contained symmetric repeats with two intact IRs. Within the Coccolithophyceae, two Phaeocystales species (*Phaeocystis antarctica* and *P. globosa*) contained asymmetric repeats, i.e., the IRa had an intact ribosomal repeat, but the IRb lacked two tRNAs [31]. In the case of *Chrysochromulina tobinii* and *Chrysochromulina parva* (Prymnesidales), the two ribosomal repeats had only one tRNA each (*tmL* in Ira and *trnA* in IRb) [35]. Two strains of *Emiliania huxleyi* (Isochrysidales) had two IRs; however, they had the shortest SSC between two IRs among all haptophyte plastid genomes [30]. In *Tisochrysis lutea* and *Isochrysis galbana* (Isochrysidales), two ribosomal repeats were aligned in same direction [31,32], and the 5S rRNA was not found in the *T. lutea* plastid genome. Finally, the two *O. neapolitana* strains had a canonical quadripartite plastid genome structure (i.e., two IRs with two tRNAs, SSC, and LSC).

### 2.3. Distribution of Genes in the Haptophyte Plastid Genomes

The majority of the genes in the haptophyte plastid genomes was conserved, but some differences in gene content were observed (Figure 5 and Appendix A). For example, 97 genes involved with photosynthesis, DNA synthesis, DNA repair, protein export, and sulfur-related genes were well conserved in the haptophyte plastid gene inventories. Three LIPOR genes of *chlB, chlL*, and *chlN* were only present in the two *O. neapolitana* plastids, whereas the *psaK* and *psbX* genes were absent in *Ochrosphaera*. The *secG* and *ycf47*, protein-export-related genes, and membrane translocator *ycf*80 gene, were absent in two Pavlovales strains. The *thiS* and *thiG* genes were absent in species of Pavlovales, Pavlomulinales, and Phaeocystales. No horizontal gene transfer (HGT) candidates were found in the plastid genome except for *rpl34* gene, which is a well-known bacterial gene transfer case to the ancestor of the haptophyte and cryptophyte plastid genomes [36]. 

Light-independent protochlorophyllide oxidoreductases (LIPOR; *chlL, chlN,* and *chlB*) are a group of enzymes that convert protochlorophyllide to chlorophyllide irrespective of the presence of light. Different from other red algal plastid descendants including cryptophytes (in pseudogenized form), alveolates, and stramenopiles, LIPOR genes have not been previously reported for haptophytes; therefore, it has been suggested that haptophytes lost these genes secondarily [27,28,29]. Here, we unexpectedly found the LIPOR genes in the plastid genomes of two *O. neapolitana* strains. To test the presence of these genes in other haptophyte taxa, we searched for the *chlL, chlN,* and *chlB* genes in 11 selected haptophyte strains using specific PCR primers. Although this taxon sampling overrepresented three haptophyte genera with well-established benthic stages (i.e., four *Chrysotila,* three *Ruttnera,* and three *Ochrosphaera* strains), we included taxa from all seven orders of the Haptophyta. Consistent with the plastid genome data, the *chlL* and *chlN* genes were found in all *Ochrosphaera* and *Chrysotila* strains (Coccolithales) and in three *Ruttnera* strains (Isochrysidales). The *chlB* gene was absent in *Chrysotila stipitata* strain A13,147 and *Ruttnera lamellosa* strain A12,715 (Appendix A). The presence of LIPOR genes in the *Ruttnera* strains is interesting, because other Isochrysidales (two *Emiliania huxleyi* strains, *Isochrysis galbana,* and *Tisochrysis lutea*) lacked the three LIPOR genes. A common feature of *Ochrosphaera, Chrysotila,* and *Ruttnera* is that they have a dominant benthic palmelloid stage (e.g., [7,14], and this study). In contrast, LIPOR genes were not found in haptophyte species that had a predominantly or entirely planktonic stage: *Diacronema, Pavlomulina, Phaeocystis, Chrysochromulina,* and *Emiliania* species (Appendix A). 

The presence of LIPOR genes in some Isochrysidales and Coccolithales species suggests two possibilities. First, the genes may have been acquired during two independent horizontal gene transfer events, because all the basal taxa lack these genes. Second, ancestral genes may be retained independently in the benthic Isochrysidales and Coccolithales but lost in other haptophyte lineages. To test these two possibilities, we compiled the LIPOR gene sequences from two *O. neapolitana* strains as well as other taxa from the NCBI database. The concatenated gene tree revealed that red algae and the red algal-derived plastid-containing lineages formed a highly supported cluster (BS = 100%) (Appendix A). Individual gene trees with more taxa are consistent with the concatenated gene tree (Appendix A). This suggests that the LIPOR genes in haptophytes, cryptophytes, alveolates, and stramenopiles were inherited from a red-algal plastid ancestor [27,28,29,37]. In addition, the *chlL* and *chlN* genes are co-localized, and this structural feature is conserved in both red algae and the secondary endosymbiotic CASH linage, as well as in Viridiplantae and glaucophytes combined, the sister groups of red algae (Appendix A). This also supports the independent gene retention hypothesis, because it is unlikely to have two independent HGTs of *chlL* and *chlN* genes located side by side. 

Based on these results, independent LIPOR gene retention within three genera and frequent gene losses in other haptophyte lineages is a more preferable hypothesis than independent gene acquisitions. This postulation can be supported by the following reasons. Horizontal gene transfer to plastid genomes is rarely observed compared to transfers into the nucleus; one exceptional case is the *rpl*34 genes in haptophyte and cryptophyte plastid genomes. It has been reported that the plastid genome is highly resistant to the uptake of intracellular DNA [38]. On the other hand, the independent loss of LIPOR genes was reported from cryptophyte plastid genomes [28]. Some cryptophyte species (e.g., *Cryptomonas curvata* and *Storeatula* sp. CCMP 1868) possess three LIPOR genes, some taxa (e.g., *Rhodomonas salina*, *Chroomonas placoidea*, and *Chroomonas mesostigmatica*) have pseudogenized genes, while other taxa (e.g., *Guillardia theta*, *Teleaulax amphioxeia*, and *Cryptomonas paramecium*) have completely lost them. Based on this finding, Kim and colleagues [28] suggested that LIPOR genes are undergoing deletion in cryptophytes. 

It is too early to discuss why some haptophytes retain LIPOR genes, but we cautiously postulate a correlation of LIPOR genes and benthic habitats. The biosynthesis of the LIPOR protein, containing the Fe-S cluster, could be metabolically disadvantageous to algae in iron-depleted ocean environments [27,39,40]. Moreover, LIPOR enzymes, similar to the nitrogenases from which they evolved, are very sensitive to oxygen, perhaps explaining why their genes were lost from algae living in oxygen-saturated environments (e.g., oxygenated layers of the open ocean) [24,26,27,41]. Conversely, in microbial mats, which are largely microaerophilic or anoxic, the retention of LIPOR genes is favored. Sassenhagen and Rengefors [40] found LIPOR genes in planktonic raphidophytes that are not benthic but migrate into the anoxic hypolimnion to access phosphorus. In contrast, the POR (light-dependent protochlorophyllide oxidoreductase) protein can perform the same function without the Fe–S cluster; therefore, the POR is advantageous in the surface of oxygen-rich open ocean [27]. In addition, both *por* and LIPOR genes can be expressed under the light conditions, but only LIPOR genes are expressed under dark conditions. Therefore, benthic haptophytes growing in shady habitats (e.g., under macroalgae or within algal mats) may benefit from the expression of LIPOR genes when producing chlorophylls [22,27]. Analogously, the wildtype of the green alga *Chlamydomonas reinhardtii* produces *chl*B and *chl*N proteins independent of light, but the *chl*L protein is only produced under dark or low light conditions (less than 15 μmol m^−2^ s^−1^). Interestingly, *Chlamydomonas* mutant cells cannot produce the *chl*L protein under any light conditions, and as a consequence, mutant cells change to a yellow color under dark conditions because protochlorophyllide accumulates [42]. 

## 3. Materials and Methods

### 3.1. Algal Cultures and DNA Preparation

*Ochrosphaera neapolitana* CCAC 3688 B was isolated from Gran Canaria, Canary Islands, Spain (27°59′20″ N, 15°22′22′′ W), and *O. neapolitana* A15,280 was isolated from Treasure Island, Florida, USA (27°45′51.92′′ N, 82°46′13.18′′ W). Strains NIES-1395 and NIES-1964, were received from the National Institute for Environmental Studies (NIES), and strain UTEX LB 1722 was obtained from the UTEX culture collection. *Diacronema* sp. strain A13,432 was collected from Thayer’s Lake, Michigan, USA (47°17′28.68′′ N, 88°15′15.48′′ W). Haptophyte strains used in a previous study [14] were also investigated: *Chrysotila stipitata* A13,112, A12,964, A13,147, and A13,110 and *Ruttnera lamellosa* A12,715. The marine strains were cultivated in 2× L1 medium with pH 7.8~8.0 [43]. The freshwater *Diacronema* sp. A13,432 was cultivated in the DY-V medium with pH 6.8~7.0 [44]. In addition, deep frozen cells of *Ruttnera lamellosa* A13,109 and A12,964 were used (see [14]). The strains used in our experiment are listed in the Appendix A.

A modified 2× CTAB method was used to extract DNA from the 14 haptophyte strains. Briefly, samples were plunged directly into liquid nitrogen for 1 min and thawed at 96 °C for 1 min. This freeze–thaw cycle was repeated three times, and the samples were then ground using a sterile micropestle. After this homogenization step, a CTAB DNA extraction method was followed using a 2× CTAB lysis solution as described in Stewart (1997) [45].

### 3.2. Species Identification and Phylogenetic Analysis

For species identification, partial 18S rRNA was amplified from 11 haptophyte strains, whereas an 18S rRNA sequence of *Chrysotila stipitata* A13110 was downloaded from NCBI (Accession number: KF696663.1), and sequences for *O. neapolitana* A15,280 and CCAC 3688 B were acquired from nuclear genome data (Appendix A). PCR primers for hapto18S-337F (CTACCATGGCGTTAACGGGT) and hapto18S-1423R (TTGCCGCAAACTTCCACTTG) were newly designed. The PCR conditions were an initial denaturing phase at 95 °C for 2 min, 30 repetitions at 95 °C for 20 s, annealing of each primer set at 58 °C for 40 s, and extension at 72 °C for 1 min. The PCR products were purified using a LaboPassTM Gel Extraction Kit (Cosmo GeneTech Inc., Seoul, Republic of Korea) and then sent for Sanger sequencing (Macrogen Inc., Seoul, Republic of Korea). 

Partial 18S rRNA sequences from PCR were used as queries for the BLASTn search (*e*-value ≤  1 × 10^−5^) to collect the homologous sequence for an alignment using MAFFT (V.8.3.10) with default option for the total of 77 haptophyte taxa. Geneious Prime (V.2020.2.4) was used for manual sequence trimming. The maximum likelihood phylogenetic analysis was conducted using IQ-TREE (V.1.6.8) with 1000 replications for the ultrafast bootstrap analysis [46]. The best evolutionary model was selected with the IQ-TREE basic option for model selection function, and a TN+F+I+G4 model for 18S rRNA and a GTR+F+G4 model for 28S rRNA were chosen as the best models, respectively. The 28S rRNA sequences for *O. neapolitana* CCAC 3688 B and A15,280 were obtained from their genome sequencing data, and other 28 rRNA sequences were downloaded from NCBI (Appendix A).

### 3.3. Morphological Observations: Light Microscopy and Scanning Electron Microscopy 

The cell morphology of four *O. neapolitana* strains (CCAC 3688 B, A15,280, NIES-1395, and NIES-1964) was observed using inverted and upright compound microscopes (Leica DMI3000, Leica Camera AG, Wetzlar, Germany; Olympus BX53, Olympus Corporation, Tokyo, Japan). The scale morphology was observed with a scanning electron microscope (SEM) using cultured cells that were fixed for 20 min with a few drops of 2% (*v*/*v*) osmium tetroxide dissolved in the 2× L1 medium. After fixation and centrifugation, the pellet was rinsed with autoclaved distilled water (repeated three times), and the cells were transferred to SEM specimen stubs covered with aluminum foil. Specimens were placed into a 65 °C dry oven for one day and then kept in desiccators filled with silica gel for three days. Dried specimens were coated with iridium and observed with a JEOL JSM-6700F Scanning Electron Microscope (JEOL Ltd., Tokyo, Japan) located at the Cooperative Center for Research Facilities, Sungkyunkwan University (Suwon, Republic of Korea).

### 3.4. Plastid Genome Construction

Genome sequencing was conducted using the Illumina platform (Novaseq 6000, DNA-Link Inc., Seoul, Republic of Korea) that generated 34 Gb raw read data. The plastid genome of *O. neapolitana* CCAC 3688 B was assembled using NOVOplasty (V.4.2) that resulted in two contigs (62 kb, 55 kb). To connect the contigs, manual PCRs were conducted. One complete circular plastid genome of CCAC 3688 B was used to assemble that of *O. neapolitana* A15,280 using NOVOplasty [47] from 46 Gb Illumina raw data. The *Emiliana huxleyi* plastid genome (NC_007288.1) was used as the reference genome for gene annotation with the programs of Geseq [48] (https://chlorobox.mpimp-golm.mpg.de/geseq.html, accessed on 10 September 2021), Aragorn (V.1.2.38), trnascane-SE (V.2.0.7), and ARWEN (V.1.2.3). Aragorn [49] (http://www.ansikte.se/ARAGORN/, accessed on 10 September 2021) and RNAmmer [50] (http://www.cbs.dtu.dk/services/RNAmmer/, accessed on 10 September 2021) were used to find tRNA and rRNA, respectively. Each predicted gene was also confirmed manually by using Geneious Prime (V.2020.2.4) and BLASTp (*e*-value < 1 × 10^−5^). All the intergenic regions (IGRs) were searched using BLASTn to identify unpredicted genes. Finally, two complete plastid genomes were visualized using OrganellarGenomeDRAW [51] [https://chlorobox.mpimp-golm.mpg.de/OGDraw.html (accessed on 10 September 2021)], Clinker (V.0.0.24) [52] and Adobe Illustrator (V.27.2).

### 3.5. Validation of the Gains and Losses of Genes

Publicly available haptophyte plastid genomes were retrieved from NCBI (Appendix A), and protein sequences were extracted to map gene gains or losses. Orthologous gene families (OGFs) of haptophyte proteins were clustered using OrthoFinder (v1.1.8). Each orthologous protein was confirmed by phylogenetic analysis and an NCBI batch CD search. For phylogenetic analyses, each individual gene was searched using BLASTp against the NCBI RefSeq (CXV_ver_201606) database. These homologous clusters were aligned using Clustal Omega (v.1.2.1). In total, 88 plastid genes were concatenated (41,508 amino acid sequences), and the alignment was manually confirmed. Phylogenetic analysis of the concatenated sequences was carried out using IQ-TRE E with LG+F+I+G4 as the best evolutionary model.

To validate LIPOR genes (*chlB*, *chlL*, and *chlN*), PCR was conducted in 12 haptophyte strains (*C. stipitata* A12,964, A13,147, A13,110, and A13,112, *R. lamellosa* A12,715, A13109, and 13130, *Isochrysis* sp. A12,896, *Diacronema* sp. A13,432, and *Ochrosphaera neapolitana* NIES-1964, NIES-1395, and UTEX LB-1722). Newly designed primers chlB-350F (CTGATGTTTTACTTGCTGATGT), chlB-1470R (ATTTAGTTCTTGTAACGCGTTT), chlN-349F (GACCTTGAGTCAATTGCAATAA), chlN-1167R (GACCTTGAGTCAATTGCAATAA), chlL-369F (ACTCGGGGATGTTGTATGCG), and chlL-850R (CACCCACAGGTGCTGATTCT) were applied for the PCR. The PCR conditions and Sanger sequencing were the same as described above. The new sequences were deposited in GenBank (Appendix A). LIPOR genes were searched from eight available genomes including the two newly generated *Ochrosphaera neapolitana* genomes. LIPOR protein sequences were acquired from NCBI, and the chlB-chlL-chlN protein sequences were concatenated (1926 amino acid sequences) and aligned using Clustal Omega (v.1.2.1). Phylogenetic analysis of these concatenated sequences was conducted with IQ-TREE using the LG+F+I+G4 model.

## 4. Conclusions

This study provided complete plastid genomes for two *O. neapolitana* strains, which represent the first two genome reports for the Coccolithales. Due to large intergenic regions, the two plastid genomes had the largest size among the published haptophyte plastid genomes. These two plastid genomes share little of the canonical ribosomal repeat structure with other haptophyte plastid genomes. Meanwhile, these two plastid genomes show infrastructural variations including the additional *ycf55* and one *trnL-UAG* in CCAC 3688 B, along with incomplete coccolith formation in A15,280 strain. LIPOR genes were retained in some haptophyte species that have a dominantly benthic form suggesting a correlation to the benthic life form (e.g., anoxic conditions in microbial mats and dark conditions under the macroalgae). Further study is needed to test this hypothesis.

## Figures and Tables

**Figure 1 ijms-24-10485-f001:**
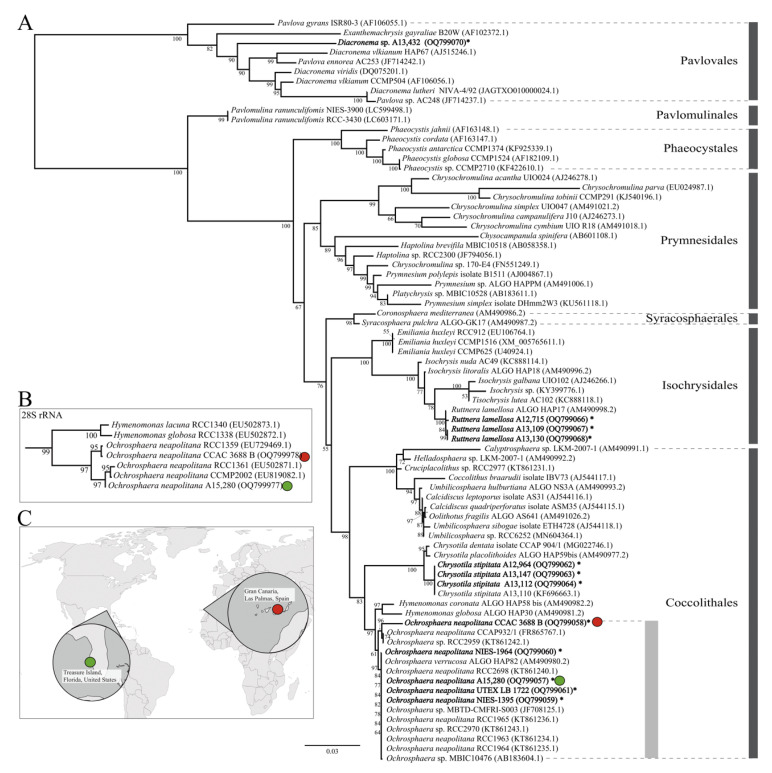
Isolation sites of two *Ochrosphaera neapolitana* strains and phylogenetic relationships inferred from 18S and 28S rRNA sequences. The 18S rRNA-based phylogeny of haptophytes. Red and green circles indicate newly isolated *Ochrosphaera neapolitana* strains, and bolds and asterisks indicate newly acquired partial 18S rRNA by PCR (bootstrap values < 50 were not presented) (**A**). Simplified 28S rRNA phylogeny from the original Appendix A. Both the 18S and 28S rRNA trees show the sister-group relationship of the genera *Hymenomonas* and *Ochrosphaera* (**B**). Isolation sites for two *Ochrosphaera neapolitana* strains from Gran Canaria, Spain, (CCAC 3688 B, Red circle) and Treasure Island, Florida, USA (A15,280, Green circle) (**C**).

**Figure 2 ijms-24-10485-f002:**
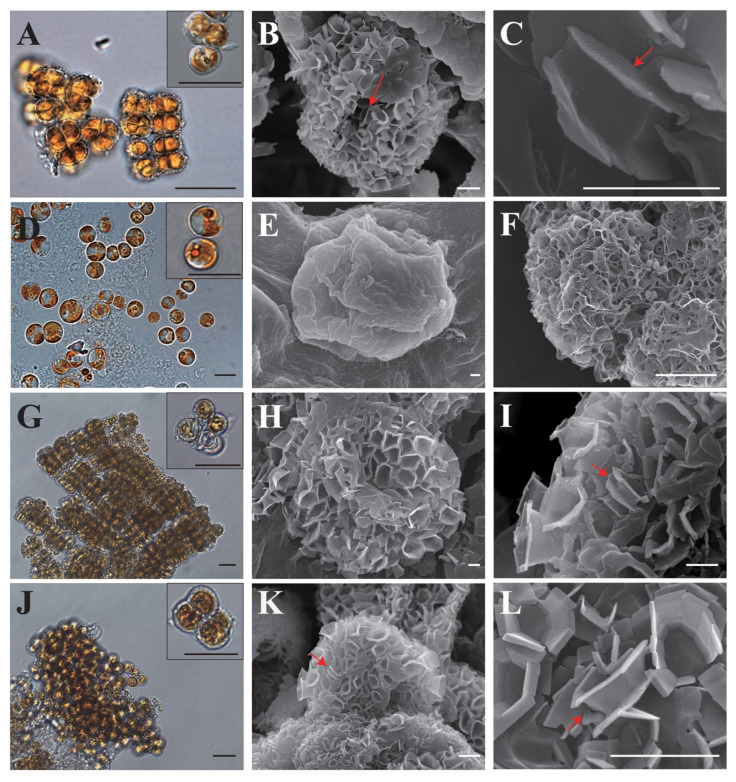
Light microscope and scanning electron microscopy (SEM) observation of *Ochrosphaera neapolitana*. Light microscopic cell and coccolith images of *Ochrosphaera neapolitana* strains. *O. neapolitana* CCAC 3688 B (**A**–**C**), NIES-1395 (**G**–**I**), and NIES-1964 (**J**–**L**) have similar colony-forming patterns and coccolith morphology, but A15,280 (**D**–**F**) shows different colony-forming patterns and the cells are naked or with an incomplete coccolith cover with a lot of holes. The red arrows indicate pully-shaped coccoliths. Scale bar = 5 µm for (**A**,**D**,**G**,**J**); 1 µm for (**B**,**E**,**F**,**H**,**I**,**K**,**L**); and 500 nm for (**C**).

**Figure 3 ijms-24-10485-f003:**
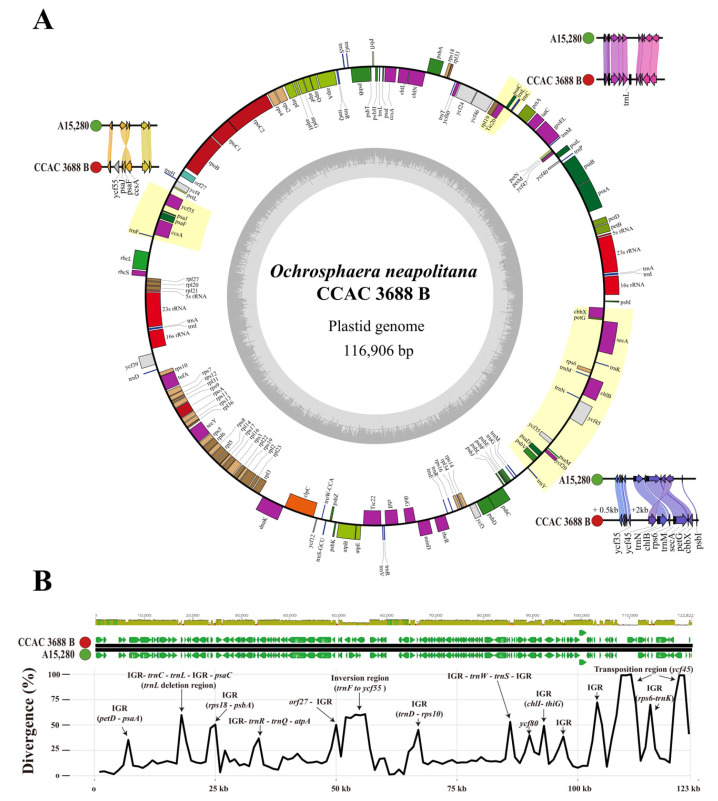
Comparison of the plastid genomes of *O. neapolitana* CCAC 3688 B and A15,280. Circular map of the plastid genomes of genomes of *O. neapolitana* CCAC 3688 B and A15,280 (**A**). Locations of additional genes and the translocation of gene synteny are illustrated beside the reference genome of CCAC 3688 B. Sequence divergence comparison between the two plastid genomes (**B**). Genes or (inter)genic regions (IGR) with high divergence rates are indicated with arrows.

**Figure 4 ijms-24-10485-f004:**
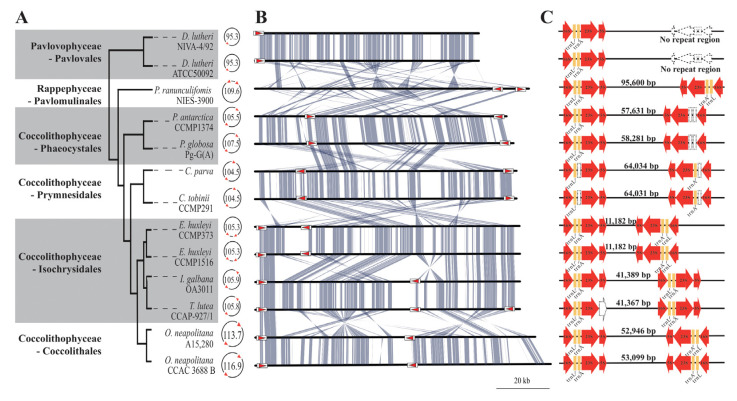
Comparison of the haptophyte plastid genome structures. Simplified phylogenetic tree and plastid sizes of 13 haptophytes (**A**). Syntenic comparison of linear chromosomal maps (**B**) and a canonical structure of the ribosomal operon repeat modalized 16S rRNA, tRNA-Ile, tRNA-Ala, 23S rRNA, and 5S rRNA (**C**). The Pavlovophyceae shows one ribosomal operon, whereas the Rappephyceae and Coccolithophyceae contain two copies of the ribosomal operon in different intergenic size and direction. The simplified phylogenetic tree was based on concatenated plastid genes (Appendix A).

**Figure 5 ijms-24-10485-f005:**
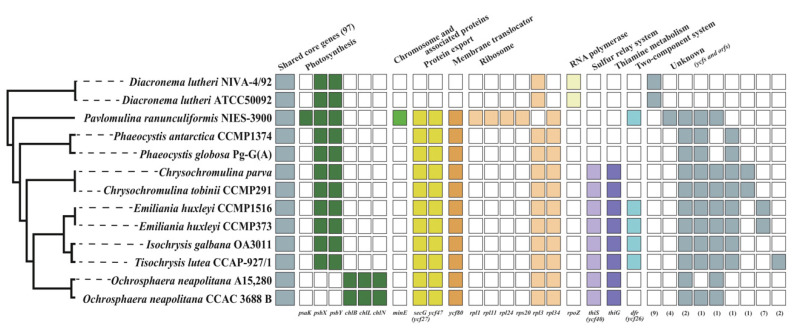
Distribution of gene contents in the haptophyte plastid genomes. Genes found in all haptophyte plastomes are collectively shown as ‘shared core genes (97)’. Color boxes indicate genes present in plastid gene inventories (right). The genes used in the heatmap were categorized by KEGG annotation as orthologous gene families, and a simplified phylogenetic tree was drawn based on the concatenated data set of 88 plastid genes (Appendix A).

**Table 1 ijms-24-10485-t001:** Comparison of 13 haptophyte plastid genomes. Newly sequenced plastid genomes of two *Ochrosphaera neapolitana* were marked in bold.

Species	Total Length (bp)	Total % GC	Ribosomal Repeat Direction	Total Gene Count	Total CDS Length	CDS (%GC)	rRNA Genes (%GC)	tRNA Genes (%GC)	IGR Average Size (bp) (% Portions in the Genome)	GenBank Accession
*Diacronema lutheri* NIVA-4/92	95,281	35.6	-	142	74,838	111 (36.7)	3 (56.7)	27 (49.1)	98.8 (14.4)	MT364382.1
*Diacronema lutheri* ATCC50092	95,281	35.6	-	142	74,838	111 (36.7)	3 (56.7)	27 (49.1)	99.5 (14.4)	KC573041.1
*Pavlomulina ranunculiformis* NIES-3900	105,550	34.7	Inverted	151	75,924	118 (34.7)	6 (48.5)	27 (49.1)	121.6 (16.8)	LC5648931.1
*Phaeocystis globosa* Pg-G(A)	107,461	35.4	Inverted	142	75,924	108 (36.3)	6 (47.9)	27 (53.6)	143.4 (18.9)	KC900889.1
*Phaeocystis antarctica* CCMP1374	105,512	35.5	Inverted	141	75,897	108 (36.7)	6 (53.5)	27 (47.9)	132.4 (17.7)	JN117275.2
*Chrysochromulina parva*	104,520	36.3	Inverted	145	77,544	112 (37.2)	6 (48.1)	27 (52.5)	112.3 (15.4)	MG520331.1
*Chrysochromulina tobinii* CCMP291	104,518	36.3	Inverted	145	77,775	112 (37.2)	6 (48.1)	27 (52.5)	110.7 (15.1)	KJ201907.2
*Emiliania huxleyi* CCMP1516	105,309	36.8	Inverted	155	80,259	119 (37.3)	6 (48.2)	30 (51.6)	89.4 (13.2)	NC_007288.1
*Emiliania huxleyi* CCMP373	105,309	36.8	Inverted	155	80,259	119 (37.3)	6 (48.2)	30 (51.8)	104.7 (14.5)	AY741371.1
*Isochrysis galbana* OA3011	105,872	36.2	Direct	147	78,903	112 (37.0)	6 (56.5)	29 (53.6)	108.8 (15.1)	MT304829.1
*Tisochrysis lutea* CCAP-927/1	105,837	36.2	Direct	144	78,900	110 (37.0)	5(48.1)	29 (54.1)	122.3 (16.3)	NC_040291.1
*Ochrosphaera neapolitana* A15,280	113,686	36.9	Inverted	143	78,735	109 (38.1)	6 (44.4)	30 (51.6)	167.6 (21.1)	OR148362
*Ochrosphaera neapolitana* CCAC 3688 B	116,906	36.9	Inverted	146	79,050	111 (38.1)	6 (44.4)	30 (51.6)	176.1 (22.0)	OR148461

## Data Availability

Two new plastid genomes are available in GenBank (accession numbers: OR148361-OR148462).

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
