# Peer review of "Plastid Genome Evolution of Two Colony-Forming Benthic Ochrosphaera neapolitana Strains (Coccolithales, Haptophyta)"

_ijms, 2023, doi:10.3390/ijms241310485_

Round 1
Reviewer 1 Report
The submitted manuscript mainly reports on two new chloroplast genome assemblies for two coccolitophore Ochrosphaera neapolitana isolates (Haptophyta, Coccolithales). Whereas completed land plant chloroplast DNAs can hardly deserve individual publication anymore given the ease of cpDNA assemblies in the age of NGS, the vast numbers of plant plastomes and only rare surprises given their high degree of conservation, things are a little different for (the diversity of) algae.
As such, more knowledge on organisms like the coccolitophores given their role in carbon cycling and as a mineral carbon sink certainly deserve all the more attention in our unfortunate times of climate change. Notably, the authors for example highlight the presence of chl genes encoding LIPOR (light-independent protochlorophyllide oxidoreductase), an issue that they also speculate on in connection to different life-styles and habitats. Altogether, the paper reads well and clear and pays much attention to details. Although the results do not include any very particularly exciting new molecular insights, the manuscript nicely combines the cpDNA assemblies with light and electron scanning microscopic, morphological and molecular phylogenetic studies and its publication could bring more attention to this interesting group of organisms.
Some minor points that I noted:
1. The geographic coordinates for Thayer’s lake rather seem to point to the middle of the Atlantic Ocean?
2. The sentence in line 119 “… using IQ-TREE 119 (V.1.6.8) with 1000 replications” should be reworded. The authors likely wish to refer to 1,000 (fast?) bootstrap replicates?
3. Is chlL-850F in line 172 supposed to read R for reverse primer?
4. trnL should be given with its anticodon for clarity since up to four different leucine-accepting tRNAs have been identified in organelle genomes before.
5. Inner grey circles in figure 3 should either be explained in the legend or this display option should be switched off in OGDRAW for figure creation.
6. The state of intron presence in the new cpDNA assemblies remained unclear to me and one couldn’t check in the new database entries, as they are not yet indicated in table 1. This is unfortunate since proper feature annotations is a crucial point for organelle DNA accessions.
7. If present, introns should be displayed in figure 3 and intron variability should be displayed in an additional (supplementary) table or even in a figure analogous to figure 5 on gene contents. I would assume that readers would be even more interested in potential intron gains and losses rather that the extended considerations on intergenic regions that could in fact be a little shortened in the text in my opinion.
Author Response
The authors thank you for their efforts. A point-by-point response was made to each comment. The reviewers made valuable suggestions that greatly improved the quality of our submission.
Thank you in advance for your kind consideration.

Reviewer 2 Report
This work exhibits exceptional quality and impeccable writing. It delves into a relatively unexplored topic, making it of significant interest. However, the manuscript needs minor revisions prior to be accepted.
Minor comments:
· Line 10, add "4" at the beginning of the line: 4 Central Collection of Algal Cultures (CCAC)…
· Line 16, replace "for" with "of": “genomes for two strains” “genomes of two strains”.
· In section "2.1. Algal Cultures and DNA preparations," include a table with the strains used to enhance readability and facilitate subsequent organization
· Line 106, the information is not clear, it seems that 11 random haptophyte taxa are used. It would be the 11 selected haptophyte strains.
· Line 127, why is the strain UTEX LB 1722 not included in the morphological observation?
· Section 2.2. Species identification and phylogenetic analysis, it would be convenient to include how the phylogenetic analysis with the 77 taxa mentioned in section 3.1 is carried out in the results.
Author Response

(The authors gave the same response as above.)
